# Microstructural Study of Two-Dimensional Organic-Inorganic Hybrid Perovskite Nanosheet Degradation under Illumination

**DOI:** 10.3390/nano9050722

**Published:** 2019-05-10

**Authors:** Lingfang Nie, Xiaoxing Ke, Manling Sui

**Affiliations:** Beijing Key Lab of Microstructure and Property of Advanced Solid Material, Institute of Microstructure and Properties of Advanced Materials, Beijing University of Technology, Beijing 100124, China; nielingfang1210@163.com (L.N.); mlsui@bjut.edu.cn (M.S.)

**Keywords:** 2D perovskite, photo-stability, TEM, microstructural analysis, degradation mechanism

## Abstract

Two-dimensional (2D) organic-inorganic hybrid perovskite materials have received substantial attention because of their exceptional optoelectronic properties. Although the applications of 2D perovskite nanosheets are promising in various optoelectronic devices, which all face harsh working conditions of light exposure, little is known about the photo-stability and degradation mechanisms of these 2D perovskite nanosheets. In this work, degradation of (C_4_H_9_NH_3_)_2_PbBr_4_ (BA_2_PbBr_4_) nanosheets when exposed to ultraviolet (UV) light and white light is explored. The morphology, optical properties, and microstructure of the nanosheets, under different conditions of light exposure, was studied in detail. UV light is more destructive compared to white light, which both led to a nanosheet breakdown. A combination of transmission electron microscopy (TEM) imaging and electron diffraction revealed that the organic moieties are most sensitive to light exposure and partial disorder toward complete disorder takes place during light exposure. Moreover, excessive light exposure further causes a [PbBr_6_]^4−^ octahedron tilt and re-ordering within the perovskite structure. This study could enrich the understanding of 2D perovskite nanosheets and their photostability, offer a new perspective in interpreting the light–perovskite interaction, and further help the design of robust and light-tunable 2D perovskite-based optoelectronic devices.

## 1. Introduction

Organic-inorganic hybrid perovskite materials have received substantial attention because of their exceptional photovoltaic properties and optoelectronic properties. Recently developed organic-inorganic hybrid perovskite solar cells based on (CH_3_NH_3_)PbI_3_ have achieved a high-power conversion efficiency (PCE) of 24.2%, from 3.81% in just a few years [1,2,3,4]. Combining high PCE with low-cost fabrication, the emerging perovskite solar cells are considered to hold great promise in scaling up for industrial use [5,6]. In conventional perovskite solar cells, perovskite materials are grown as polycrystalline thin films [7,8,9] or single crystals [10], where the size of the perovskite material ranges from hundreds of nanometers to millimeters [11]. In contrast, another category of hybrid perovskite materials, two-dimensional (2D) organic-inorganic hybrid perovskite nanosheets characterized by their layered structure along the *c*-axis and a large diameter-to-thickness ratio, has recently been developed and investigated [12,13,14]. Compared with three-dimensional (3D) perovskites, 2D perovskite nanosheets were shown to have a high absorption coefficient, ambipolar charge transport, long exciton diffusion length, low trap density, and low binding energy of exciton [8,11,15,16,17,18,19,20,21]. Unlike 3D perovskites, 2D perovskite nanosheets are much more stable under the presence of moisture and have been doped into 3D perovskite solar cells to improve their moisture tolerance [22]. These properties lead to potential applications of 2D perovskite nanosheets in electronic devices, such as solar cells, light-emitting diodes, photodetectors, ultrafast photonic devices, and field-effect transistors [23,24,25,26,27,28].

Generally, 2D perovskites have a natural quantum well structure [29,30], which is formed by a sandwiched configuration of corner-sharing [PbX_6_]^4−^ octahedra between organic ammonium ion barrier layers, where X represents halides, including Cl^−^, Br^−^, and I^−^. The regular formula of 2D organic-inorganic hybrid perovskite nanosheets is L_2_[ABX_3_]*_n_*_−1_BX_4_, where L represents a large cation (mostly large size or long chain organic cations), A represents a regular cation, such as Cs^+^ or methylammonium (MA^+^), B represents a divalent metal cation, such as Pb^2+^ or Sn^2+^, and (*n* − 1) is the number of perovskite unit cells. The layer spacing between the perovskite layers can be adjusted by changing the size of L and A [31]. Chen et al. obtained ultrathin and large-sized (C_4_H_9_NH_3_)_2_PbBr_4_ cross-stars with a high yield and good reproducibility by tuning the solvent volume ratio, crystallization temperature, and solvent polarity [32]. Weidman et al. demonstrated that variations of *n*, B, and X could lead to large changes in the absorption and emission energy, while variation of the A cation could lead to subtle changes, but significantly impact the nanosheets stability and photoluminescence (PL) quantum yield [20].

Although much research has focused on the synthesis and property tuning via different strategies, the photo-stability of the 2D perovskite nanosheets was not yet fully investigated. Weidman et al. noticed that high intensity of UV light illumination (~5–10 mW) could cause nanosheets to transform into thicker and bulk-like particles [20]. Wei and coworkers found that the 2D perovskite luminescent framework tended to be destroyed in a photo-chemical reaction where both the inorganic species and the organic moieties play an important role [33]. They further improved the photostability by introducing fluorine atoms on the phenyl ring of the amine in the phenylethylamine-based 2D perovskites [34]. However, little is known about the degradation mechanism of 2D perovskite nanosheets and the stability of 2D perovskite nanosheets in the presence of UV light and white light remains unclear. Since the applications of 2D perovskite nanosheets have shown to be promising in various optoelectronic devices, which all face harsh working conditions of light exposure, further studies into the photo-stability of 2D perovskite nanosheets are essential.

Transmission electron microscopy (TEM) is one of the most straightforward characterization methods in revealing the microstructures of materials and has helped with understanding the relation between structural evolution and property change in various materials. However, a limited amount of work has been reported on organic-inorganic hybrid halide materials due to their beam-sensitivity when exposed to electron beams [35,36]. By minimizing electron doses for bulk 3D perovskite and thin film perovskite, important progress has been made recently on MAPbI_3_, where twin structures, self-organized superlattice, and phase coexistence, etc., are reported [37,38,39]. On the other hand, TEM on 2D perovskites remain challenging as they are even more sensitive to electron beams. Immediate degradation of nanosheets and nanoparticles precipitation was noticed, even when imaging conditions were carefully chosen, preventing further detailed studies into 2D perovskite structures [14,21,40,41].

In this work, the degradation of BA_2_PbBr_4_ nanosheets, when exposed to either UV light or white light, was studied. The morphology, crystallinity, optical properties, and microstructures of the nanosheets, under different conditions of light exposure, were studied by X-ray diffraction (XRD), photoluminescence emission, UV-Vis absorption, and particularly TEM. By combining TEM imaging of the nanosheet morphology with electron diffraction, fine details of the structural transformation are revealed and different degradation mechanisms introduced by UV light or white light are discussed.

## 2. Materials and Methods

### 2.1. Materials Development

PbBr_2_ (99.5%) and BABr (99.5%) were purchased from Xi’an Polymer Light Technology Corp (Xi’an, China) and stored in a glovebox filled with nitrogen. Toluene was purchased from Sinopharm Chemical Reagent Co., Ltd (Beijing, China). *N*,*N*-dimethylformamide (DMF, 99.9%) was purchased from Aladdin (Beijing, China). All chemicals were used as received without additional purification.

### 2.2. Synthesis of 2D (BA)_2_PbBr_4_ Nanosheets

All solution preparation and nanosheets growth processes were carried out in ambient conditions. First, 0.1 M PbBr_2_ and 0.1 M BABr in DMF were prepared by dissolving 18.45 mg of PbBr_2_ in 0.5 mL of DMF and 15.2 mg of BABr in 1 mL of DMF, separately. The solutions were sonicated for 20 min and then stirred during heating at 70 °C for 2 h. PbBr_2_ and BABr were then mixed in a stoichiometric ratio of 1:2 and continued to be stir-heated for one hour. Finally, 10 μL of the as-prepared precursor solution was dropped into 10 mL of toluene solution during rapid stirring at room temperature and 2D nanosheets were instantaneously produced, as the color of the solution turned from transparent to blue-purple.

### 2.3. Characterization

TEM was performed on an FEI Tecnai T20 (FEI, USA) operated at 200 kV. TEM samples were prepared by dropping the nanosheet solution onto a carbon film supported Cu grid. Morphology of the nanosheets was acquired using bright field imaging. Spot size 3 was used to minimize the electron dose, due to the beam sensitivity of the 2D nanosheets.

Powder X-ray diffraction (XRD) analysis of the as-prepared materials was carried out on a Bruker D8 Advance X-ray diffractometer with CuKα radiation (λ = 1.5439 Å) (Bruker, Karlsruhe, Germany). The sample was prepared using 0.5 M precursor stock solutions. The resultant colloidal solution of nanosheets was then centrifuged at 5,000 rpm for 5 min, re-dispersed in a small quantity of toluene (~200 μL), and drop-cast onto silicon slides.

PL spectra and UV-Vis spectra were recorded using a HITACHI F-7000 fluorescence spectrometer (Hitachi, Tokyo, Japan) and a UV 2600 UV/Vis spectrophotometer (Shimadzu, Shanghai, China) at room temperature, respectively. Samples were excited by a 320 nm Xenon lamp (Hitachi, Tokyo, Japan) for photoluminescence measurements.

Atomic force microscopy (AFM) measurements were performed on an AGILENT 5400 (Agilent, Palo Alto, CA, USA). Nanosheets were dispersed on Si substrates. Tapping mode was used for imaging. Height images were acquired by both forward and reverse scan directions.

## 3. Results and Discussion

The morphology and optical properties of the as-synthesized BA_2_PbBr_4_ nanosheets were studied, as shown in Figure 1. The nanosheets show a square-shaped or rectangular-shaped morphology, as revealed by TEM (Figure 1a). The facetted shape of nanosheets indicates that the as-synthesized nanosheets have a high crystallinity. The selected area electron diffraction (SAED) pattern obtained from the nanosheet is shown as an inset in Figure 1a. Sharp diffraction spots indicate a high crystallinity of the as-synthesized nanosheets and the pattern can be indexed using the structure BA_2_PbBr_4_, confirming the successful synthesis of BA_2_PbBr_4_ nanosheets. As demonstrated in Figure 1a, the nanosheet is viewed along the [001] zone axis, whereas the side exposed facets are the crystallographic {110} planes. The side length distribution of the as-synthesized nanosheets was obtained by measuring more than 100 different nanosheets from different TEM images, as plotted in Figure 1b. It can be seen that most nanosheets range from 0.5 μm to 1.0 μm; a few nanosheets even grow as large as 2.5 μm. The thickness of the as-synthesized nanosheets was measured using AFM, as shown in Figure 1c, indicating a homogeneous distribution from 35 nm to 70 nm. The optical properties of the as-synthesized nanosheets were characterized by UV-Vis absorbance spectra (black line) and PL spectra (red line), as shown in Figure 1d. It can be seen that the absorption peak has a wavelength of 404 nm, while the PL peak has a wavelength of 411.4 nm. The Stokes shift, of approximately 8 nm, implies that the photoluminescence of the nanosheets is generated from a direct exciton recombination process. The emission has a small full-width at half-maximum of 11.45 nm, suggesting a high purity of the products. The band gap of the BA_2_PbBr_4_ nanosheets was calculated to be 3.01 eV from the UV absorption spectra.

In order to study the degradation of BA_2_PbBr_4_ nanosheets when exposed to light, the as-synthesized nanosheets were exposed to ultraviolet (UV) light irradiation and white light irradiation for different periods of time, separately. Nanosheets were dispersed in toluene and then irradiated by light, before being characterized by XRD and PL. Figure 2a demonstrates the XRD patterns of the nanosheets irradiated by UV light for 0 min (initial state), 1 min, 3 min, and 5 min, separately. It can be seen that the (*002n*) peaks have the strongest intensity (indicated in red), consistent with the layered structure of the 2D nanosheets stacked along the ***c*** axis, i.e., the <001> direction. As the irradiation time increases, both the intensity of the characteristic peaks and the peak position remains nearly the same. Hence, no lattice expansion or reduction is observed. However, the PL spectrum of the same set of nanosheets after UV light irradiation shows a difference (Figure 2b). It can be seen that the characteristic peak at 411.4 nm is present for all 4 samples without shift, but the peak intensity decreases dramatically after irradiation. Meanwhile, a new peak at 437 nm appears after an irradiation of 3 min. For comparison, the XRD and PL spectrum of the nanosheets irradiated by white light for 0 min (initial state), 5 min, 10 min, and 30 min are shown in Figure 2c,d, respectively. The XRD spectrum shows that the (*002n*) peak intensity slightly decreases during irradiation, indicating that the layered structure is relatively well maintained. The PL spectrum shows a significant decrease of intensity at the peak position of 411.4 nm after irradiation (Figure 2d). However, no other peaks appear in the PL spectrum during irradiation, which is different from the UV irradiation. It has been reported that the PL intensity is related to defects in the 2D perovskite structure [42]. Therefore, it can be inferred from the intensity decrease of the PL spectrum that both UV light and white light irradiation can introduce defects into the nanosheets. In addition, it is also worth noticing that PbBr_2_ can be identified in some XRD measurements, as indicated by black triangles in Figure 2a,c. As there is no strong correlation with the irradiation time and considering that PbBr_2_ was used as starting material for synthesizing the 2D nanosheets, it is likely that PbBr_2_ is present as a trace material, rather than as a decomposition product due to irradiation.

Although the PL spectra demonstrates significant degradation of 2D nanosheets after irradiation by either UV light or white light, the XRD patterns do not show much difference from a structural point of view. Therefore, a detailed microstructural investigation of the 2D nanosheets at different irradiation conditions is essential. TEM was used to study a series of samples. It has to be mentioned that 2D perovskite is highly sensitive to electron beam irradiation [14,21,40,41]. Nanosheets degraded almost immediately when we tried to take high resolution TEM (HRTEM) images (Appendix A). Nanoparticles precipitated rapidly and the nanosheets were damaged (Appendix A). Hence, in this work, we acquired TEM images at low magnification in order to avoid beam damage. Meanwhile, we used electron diffraction patterns instead of HRTEM images for structural investigation in detail.

Figure 3 demonstrates the nanosheets exposed to UV light. TEM images (Figure 3a–e) and the corresponding SAED patterns from the circled area (Figure 3f–j) were acquired from nanosheets exposed to UV light for 0, 1, 3, and 5 min, separately. The initial nanosheets without irradiation (Figure 3a,f) show a well-facetted square shape and the corresponding SAED pattern shows sharp diffraction spots, indicating a highly crystallized structure of the BA_2_PbBr_4_ nanosheets. The SAED pattern can be indexed according to the BA_2_PbBr_4_ [001] zone axis, confirming the viewing direction as the *c* axis of the 2D nanosheets. When the nanosheets are irradiated for 1 min, the edges became rough and etched, while the overall shape remains square-like, as demonstrated in Figure 3b. The SAED pattern (Figure 3g) is obtained from the central area of the nanosheet and the diffraction spots remain sharp, indicating good crystallinity. When the nanosheets are irradiated for 3 min and 5 min, respectively, the square-like morphology becomes more and more irregular (Figure 3c–e). Although some nanosheets are broken, as shown by the circled area in Figure 3c,d, the corresponding SAED patterns (Figure 3h–i) still show a single crystal structure with sharp diffraction spots that can be indexed as the [001] zone axis of BA_2_PbBr_4_. However, in Figure 3f–h some extra diffraction spots were present, as indicated by green and yellow arrows, and in Figure 3i–j some diffraction spots are absent, which implies that some changes took place in the fine structure of the nanosheets during irradiation. This will be discussed later in this paper. Meanwhile, after irradiation for 5 min, it was noticed that a considerable amount of nanosheets become thicker and stacked together, as shown in Figure 3e. The corresponding SAED however is the same as the one of the broken thin nanosheets (Figure 3j) and can be indexed as [001] BA_2_PbBr_4_ as well. A similar phenomenon has been reported earlier where L_2_[MAPbBr_3_]PbBr_4_ (L is an organic ligand including octylammonium and butylammonium, etc.) nanoplatelets transformed to a thick bulk-like structure under a constant illumination of 365 nm laser light [40]. The same paper also mentioned that, as the irradiation time increased, peaks at lower energy also appeared in the PL spectrum, which is again similar to our study (Figure 2b). Thus, we infer that when the nanosheets are exposed to UV irradiation, the emerging peak at 437 nm in the PL spectrum is likely due to the thickening of the nanosheets, where the bandgap shifts as the structure turns from 2D to 3D-like.

Complementary to the study of BA_2_PbBr_4_ nanosheets exposed to UV light, a similar study was performed on nanosheets exposed to white light for a series of time. Firstly, nanosheets in toluene antisolvent were studied by TEM, as shown in Figure 4. Figure 4a–e presents nanosheets irradiated for 0, 5, 10, and 30 min, separately. It can be seen that the square shape of the nanosheets becomes irregular with increasing irradiation, which is similar to the case of UV light. However, it is interesting to notice that, when the nanosheets are exposed to both white light or UV light for 5 min, the nanosheets exposed to white light change only slightly (Figure 4b) while the nanosheets exposed to UV light are already broken or stacked (Figure 3d,e). After 10 min of exposure to white light, the nanosheets turn to an irregular shape and, in the meantime, small holes or cracks appear inside the nanosheets, as indicated by red arrows in Figure 4c. As the irradiation time increases to 30 min, the holes become larger and the nanosheets became broken, as indicated by red arrows in Figure 4d. Nevertheless, the single crystal characteristics of the nanosheets remain almost unchanged with the irradiation time, as revealed by the corresponding SAED patterns in Figure 4e–h, respectively, which can be indexed as [001] BA_2_PbBr_4_ with sharp and bright diffraction spots. Again, the presence of extra diffraction spots in Figure 4e–g (as indicated by green and yellow arrows), imply fine structure changes in the irradiated nanosheets, which will be discussed further.

Comparing the results obtained from nanosheets under illumination of UV light and white light, the differences in the degradation process can be summarized in three aspects. First, nanosheets are more resistant to the irradiation of white light, compared to UV light. The morphology of the nanosheets changes drastically after 5 min of UV irradiation, but changes only slightly after the same time of white light irradiation. After only 30 min of white light irradiation, the nanosheets show a different morphology. Second, the degradation starts from the edge toward the center of the nanosheets during UV irradiation, whereas the white light is more likely to introduce defects simultaneously at the edge and at the interior of the nanosheets, as cracks and holes are detected by TEM. Third, at the later stage of irradiation, UV light leads to stacking of nanosheets which turns into thick bulk-like “particles”, but white light does not seem to have such effect. After white light irradiation, nanosheets show a mainly 2D structure although with an irregular shape. One possibility is that UV light alters the surface structure or bonding state of the nanosheets and makes their surface polarized or charged, which attracts nanosheets to stack together.

On the nanosheets SAED patterns, it is noticed more than once that, although the patterns can be indexed to [001] BA_2_PbBr_4_ using the major diffraction spots such as (200) and (220), fine details are not always the same. Extra diffraction spots sometimes appear and some diffraction spots sometimes distinguish, as mentioned earlier in the paper (Figure 3 and Figure 4). Therefore, we looked into a few tens of SAED patterns and the most representative diffraction patterns under each condition were selected and shown in Figure 5a–c. Figure 5a shows the most representative SAED pattern of the as-synthesized nanosheets, while Figure 5d shows a simulated diffraction pattern of [001] BA_2_PbBr_4_ for comparison. It was simulated using the original BA_2_PbBr_4_ structure, as demonstrated in Figure 5g (top view of [001]) and Figure 5j (side view of [100]). Clearly, the initial diffraction pattern (Figure 5a) matches the simulated pattern (Figure 5d) to a high degree. However, in some nanosheets, such as Figure 3f, (*2n* + *1*, *k*, *0*) and (*0*, *2n* + *1*, *0*) diffraction spots are present with very low intensity, as indicated by green arrows and yellow arrows, respectively. According to the *Pbca* space group, the (*2n* + *1*, *k*, *0*) and (*0*, *2n* + *1*, *0*) reflections should be extinct. Yet they are present, although with a weak intensity, indicative of a symmetry change in the initial structure. Since the as-synthesized nanosheets with high crystallinity are supposed to keep their initial structure, it is likely that the organic moieties deviate slightly from the perfect structure and, hence, alter the structure symmetry. In order to confirm this hypothesis, we modified the BA_2_PbBr_4_ structure by partially displacing a few organic moieties randomly away from their initial positions (as illustrated in Figure 5h,k) and then simulated the electron diffraction pattern (Figure 5e). The simulated pattern matches the experimental data very well (Figure 3f). The cause of organic moieties’ displacement might have two possibilities. First, the initial structure might not be perfect and some organic moieties might be randomly arranged. Second, incident electrons might have knocked out organic moieties, which are highly beam sensitive. It was reported earlier by Rothmann et al., that the organic-inorganic hybrid halide perovskite structure was highly sensitive to electron beam irradiation and an extremely small dose of 1e^−^/Å^2^s was used to acquire electron diffractions in order to avoid structural alternation and symmetry breaking [39]. In our experience, 2D perovskite nanosheets are even more sensitive to electron irradiation and, therefore, a limited electron dose was used. Nevertheless, it was likely that, within a few seconds of exposure to the electrons, the structure could have been altered.

Figure 5b demonstrates the nanosheets after 1 min of UV irradiation, which is found to be the same as nanosheets after 5 and 10 min of white light irradiation. It can be seen that the (*2n* + *1*, *k*, *0*) and (*0*, *2n* + *1*, *0*) reflections become more intense, compared to their initial state. Thus, it can be inferred that UV/white light irradiation introduces more defects into the 2D perovskite structure by first knocking out or misplacing organic moieties.

Furthermore, after irradiation by UV light for 3 min and white light for 30 min, the symmetry of the diffraction pattern is altered again (Figure 5c). Only the (*0*, *2n*, *0*), (*2n*, *0*, *0*), and (*2n*, *2n*, *0*) remain, compared to the original structure, whereas additional (*2n* + *1*, *2n* + *1*, *0*) reflections appear and the (*2n + 1*, *k*, *0*) as well as (*0*, *2n* + *1*, *0*) reflections start to fade. The diffraction pattern shows 4-fold symmetry. This lasts after even further irradiation by UV light for 5 min and by white light for 30 min (Figure 5c). The disappearance of (*2n* + *1*, *k*, *0*), as well as (*0*, *2n* + *1*, *0*), is due to the disorder of the organic moieties during further illumination, where the organic moieties become randomly oriented. However, they remain in the 2D nanosheets to keep the layered structure, as confirmed by the presence of the (*0*, *0*, *2n*) peaks in the XRD measurements. On the other hand, the emergence of the (*2n* + *1*, *2n* + *1*, *0*) diffraction spots is due to a superstructure along the <110> direction, which is likely caused by an ordered arrangement of the [PbBr_6_]^4–^ octahedra. Therefore, the BA_2_PbBr_4_ structure is modified by tilting the [PbBr_6_]^4–^ octahedra along the <110> direction in a zigzag manner, as illustrated in Figure 5i,l, using the [001] top view and [100] side view. In order to observe the rotation of the [PbBr_6_]^4–^ octahedra more clearly, the organic molecules between the BA_2_PbBr_4_ layers were hidden from display. The corresponding simulated electron diffraction pattern along [001] is shown in Figure 5f, which fits the experimental data. The tilting of the [PbBr_6_]^4–^ octahedra is merely one of the possibilities. Titling, rotation, or deformation of octahedra is known to change the symmetry of the perovskite structure and single crystal XRD is one of the common ways to confirm its real structure. However, due to the nature of the 2D nanosheets which tend to stack along the *c* axis, it is not possible to perform a single crystal XRD analysis at current stage. Similarly, the 2D nanosheets tend to stay flat when dispersed on a TEM grid and therefore it is not possible to tilt to other zone axes to search for eventual distinction conditions. However, the diffraction simulation offers a way of interpreting the diffraction pattern and points out that excessive light illumination is likely to cause [PbBr_6_]^4–^ octahedra tilting and re-ordering. It was reported earlier that PL emission is correlated to structural modulation, particularly to Pb-(µ-Br)-Pb angle change [42]. Therefore, lattice distortion, where [PbBr_6_]^4–^ octahedra are tilted and Pb-(µ-Br)-Pb angle is altered, may account for PL intensity decrease and property degradation. It is important to note that, during irradiation of both UV light and white light for as long as 5 and 30 min, respectively, the group of (*2n*, *0*, *0*) and (*0*, *2n*, *0*) reflections remain present. The (*2n*, *0*, *0*) and (*0*, *2n*, *0*) diffraction spots are related to Pb^2+^ and, therefore, it can be confirmed that the Pb^2+^ is robust to electron beam irradiation.

The detailed study of the SAED patterns during irradiation has further revealed information of the structural degradation. Although the perovskite structure is maintained, a removal of the organic moieties and re-ordering of the [PbBr_6_]^4–^ octahedra takes place, degrading the structure and modifying the Pb-Br bonding, which is closely related to its optoelectronic properties.

## 4. Conclusions

The degradation mechanism of BA_2_PbBr_4_ nanosheets exposed to UV light and white light was studied in detail by TEM. Although the layered structure of the nanosheets remains well conserved in both cases, the detailed microstructure is altered by UV light and white light irradiation at different speed. When the nanosheets are exposed to UV irradiation, the 2D nanosheets degrade from the edges towards the interior. After 5 min of exposure, the nanosheets are either broken or stacked together in the presence of antisolvent toluene, resulting in a significant degradation of their optical properties, as revealed by the PL spectra. When the nanosheets are exposed to white light, defects are generated at the edges and interior simultaneously. Compared to UV light, while light is less destructive and allows the structure to remain as long as 30 min. Further studies by SAED reveal a fine structural alteration by light irradiation, which first causes a partial disorder of the organic moieties within the nanosheets and then complete disorder of the organic moieties, together with a [PbBr_6_]^4–^ octahedral tilt-order, although the perovskite structure remains. The fine microstructural alteration is believed to be closely related to the deterioration of the optoelectronic properties. This study reveals a different degradation mechanism of the BA_2_PbBr_4_ nanosheets from a structural aspect and diffraction studies helped us to unveil the dynamics of the organic moieties and the [PbBr_6_]^4–^ octahedra during irradiation. We hope this study could help the community in (1) improving the photo-stability of 2D perovskite by using more robust organic moieties with stronger bonding to the [PbX_6_]^4–^ octahedral layer and (2) harnessing degradation mechanisms to tune the 2D perovskite property using light exposure as a tool. Hence, we believe this study can enrich the understanding of 2D perovskite nanosheets and their photostability and further push the design of 2D perovskite-based optoelectronic devices.

## Figures and Tables

**Figure 1 nanomaterials-09-00722-f001:**
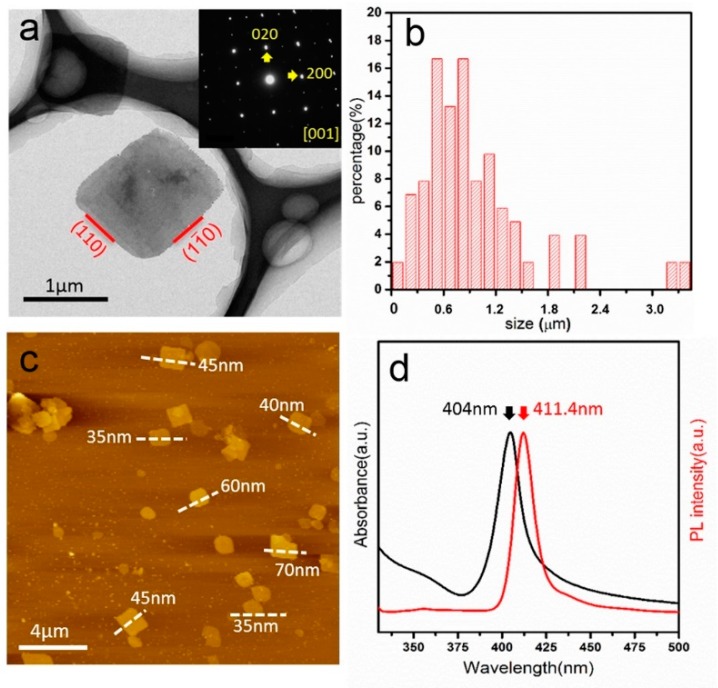
(**a**) TEM image of the as-synthesized BA_2_PbBr_4_ nanosheets along the [001] zone axis and the corresponding SAED pattern (inset). (**b**) Side length distribution of the as-synthesized BA_2_PbBr_4_ nanosheets; (**c**) AFM image of the BA_2_PbBr_4_ nanosheets with the measured thickness results; and (**d**) absorption spectrum (black line) and PL spectrum (red line) of the BA_2_PbBr_4_ nanosheets in toluene.

**Figure 2 nanomaterials-09-00722-f002:**
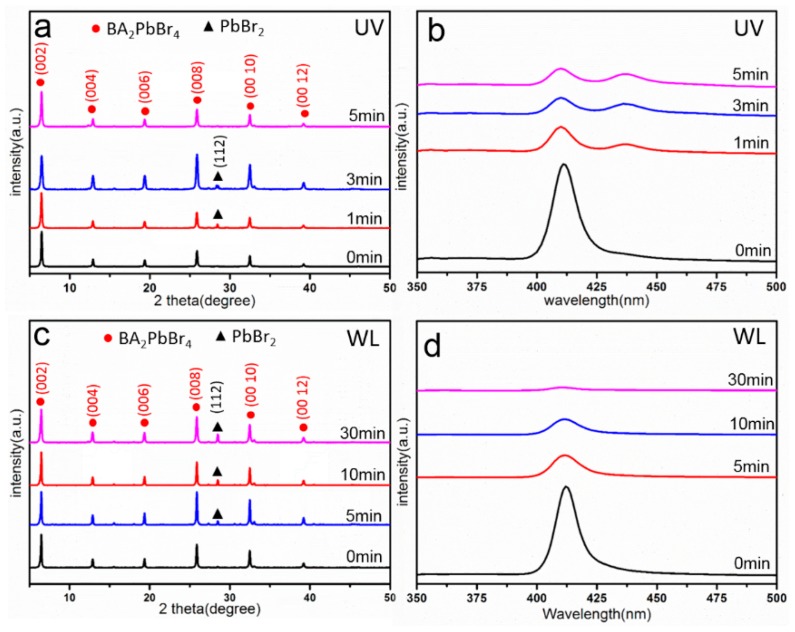
(**a**,**b**) XRD and PL spectra of BA_2_PbBr_4_ nanosheets after UV irradiation for 0, 1, 3, and 5 min. (**c**,**d**) XRD and PL spectra of BA_2_PbBr_4_ nanosheets after white light irradiation for 0, 5, 10 and 30 min.

**Figure 3 nanomaterials-09-00722-f003:**
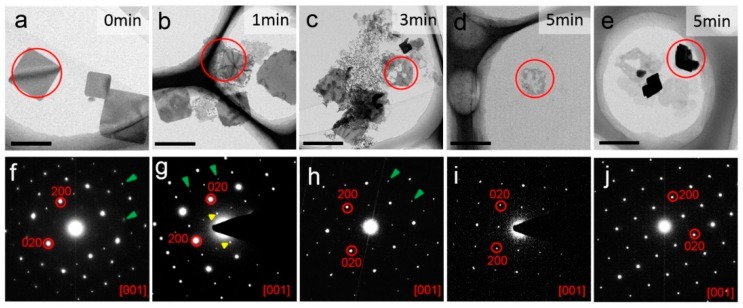
(**a**–**e**) TEM images of BA_2_PbBr_4_ nanosheets after UV irradiation for 0 min, 1 min, 3 min, and 5 min (bar = 1 μm); (**f**–**j**) The corresponding SAED patterns acquired from the circled area in (**a**–**e**), respectively. Green and yellow arrows indicate the presence of extra diffraction spots.

**Figure 4 nanomaterials-09-00722-f004:**
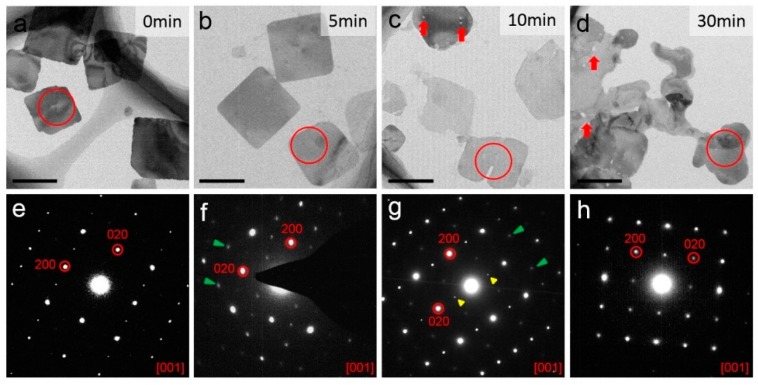
(**a**–**d**) TEM images of BA_2_PbBr_4_ nanosheets after white light irradiation for 0, 5, 10, 30 min, respectively (bar = 1 μm); (**e**–**h**) the corresponding SAED patterns of the nanosheets marked by red circles in (**a**–**d**), respectively. Green and yellow arrows indicate the presence of extra diffraction spots.

**Figure 5 nanomaterials-09-00722-f005:**
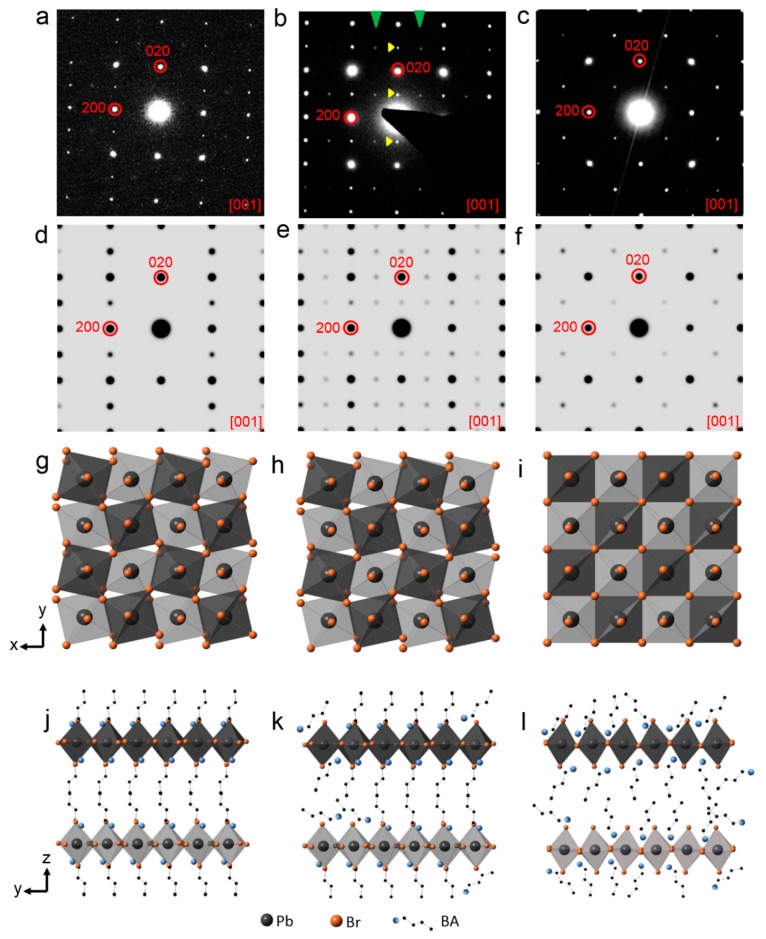
(**a**–**c**) Representative SAED patterns obtained from BA_2_PbBr_4_ nanosheets exposed to UV light or white light at the initial state, intermediate state, and excessive state, respectively. (**d**,**g**,**j**) Simulated [001] electron diffraction of the initial BA_2_PbBr_4_ structure with the corresponding illustration from [001] top view and [100] side view, respectively. (**e**,**h**,**k**) Simulated [001] electron diffraction of the BA_2_PbBr_4_ structure where organic moieties were partially disordered with the corresponding illustration from [001] top view and [100] side view, respectively. (**f**,**i**,**l**) Simulated [001] electron diffraction of the BA_2_PbBr_4_ structure where organic moieties were disordered and [PbBr_6_]^4–^ octahedra were tilt-ordered along <110>, with the corresponding illustration from [001] top view and [100] side view, respectively.

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
