# Peer review of "Microstructural Study of Two-Dimensional Organic-Inorganic Hybrid Perovskite Nanosheet Degradation under Illumination"

_nanomaterials, 2019, doi:10.3390/nano9050722_

Reviewer 1 Report

It has been attached herewith.

Author Response

Dear Reviewer,

We would like to thank you for your effort in reading through the manuscript and providing us with encouraging and helpful comments. We have made changes according to your suggestions and have listed the changes after corresponding comments. You can download the pdf "reviewer report -1 final" for  detail answers.

Please do not hesitate to let us know if there are any other questions.

Thank you very much for your comments and suggestions.

With best regards,

Lingfang Nie, Xiaoxing Ke and Manling Sui

Reviewer 2 Report

This manuscript dealing with light-induced degradation of 2D sheets of organic-inorganic hybrid perovskite materials is suitable in content for Nanomaterials, but requires revision. The English language in the paper needs significant attention before the paper can be published. The paper is difficult to read and follow at present, and the impact of the work is lost. After revision of the presention and after the minor changes below, I would like to re-review the paper.

Title: why does nanosheets have an apostrophe? This makes no sense. or should this be a colon since "Degradation.....' is a secondary title?

Abstract line 12: should be 'applications of  2D perovskite '

Abstract line 24: what does 'enlighten the design' mean?

Throughout - be consistent with the formula [PbBr6]4– or [PbX6]4–. Use [ ] as shown throughout the manuscript. In at least one place, the [ ] are missing.

p.2 line 47  'ammonium' needs to be 'ammonium ion'

p. 2 line 90 The identities of A and B need to be specified in the experimental section; 'was prepared' should be 'were prepared'

line 94. The phrase '2D nanosheets were instantaneously produced' requires a visual description to aid the experimentalist following the procedure.

Line 144. Is 'dissolved' correct or should this be 'dispersed'? 

Line 244. The phrase 'altered the surface bonding state' is not clear and needs explaining or rephrasing.

Line 264 It is unclear what the modification described as 'partially misplacing organic moieties' is. Please explain more clearly. Should 'misplacing' be 'displacing'?

Line 266. The phrase 'The cause of organic moeities.....' does not make clear sense.

Line 314. What do the authors envisage when they say 'Pb-Br bonding coujld be altered'? This is very vague.

If I am correct, the reference style requires DOI numbers.

Author Response

Dear Reviewer,

We would like to thank you for your effort in reading through the manuscript and providing us with encouraging and helpful comments. We have made changes according to your suggestions and have listed the changes after corresponding comments.

1.    Replies to your comments can be found in this report, with each suggestion or comment made by the referees addressed point by point.

2.    We have asked a native English speaker to improve the language of the manuscript, and therefore the updated manuscript is uploaded in “track change” mode. If you turn the “track change” mode on, the language improvement can be noticed.

3.    Meanwhile, content change is also included in the “track change” mode. However, in order to assist reading, the content changes to address the reviewer’s comments are highlighted in yellow in the same “track change” version.

We also uploaded a “clean” version where all editing has been accepted and only content improvement is highlighted in yellow.

4.    Meanwhile, we realized that the figures of electron diffraction pattern had mistakes in describing diffraction spots. The indexation should be in Miller index and therefore the brackets should not be present. Hence, we have removed all brackets from SAED patterns. Please see the improved manuscript.

Thank you very much for your comments and suggestions.

With best regards,

Lingfang Nie, Xiaoxing Ke and Manling Sui

Round  2

Reviewer 1 Report

The authors have revised the manuscript addressing the comments. The work is well discussed with experimental and theoretical results. 

I recommend this paper for publication.

Cheers!!!

Reviewer 2 Report

The revisions have been properly dealt with and the English is now much improved. Accept.